# A Multicenter Retrospective Cohort Study on Superior Vena Cava Resection in Non-Small-Cell Lung Cancer Surgery

**DOI:** 10.3390/cancers14246138

**Published:** 2022-12-13

**Authors:** Andrea Dell’Amore, Alessio Campisi, Luca Bertolaccini, Chunji Chen, Piotr Gabryel, Chunyu Ji, Cezary Piwkowski, Lorenzo Spaggiari, Wentao Fang, Federico Rea

**Affiliations:** 1Division of Thoracic Surgery, Department of Cardiothoracic Surgery and Vascular Sciences, Padua University Hospital, University of Padua, 35128 Padua, Italy; 2Thoracic Surgery Department, University and Hospital Trust–Ospedale Borgo Trento, 37126 Verona, Italy; 3Department of Thoracic Surgery, Shanghai Chest Hospital, Shanghai Jiao Tong University, Shanghai 200032, China; 4Department of Thoracic Surgery, IEO, European Institute of Oncology IRCCS, 20141 Milan, Italy; 5Department of Thoracic Surgery, Poznan University of Medical Sciences, 61-701 Poznan, Poland; 6Department of Oncology and Hemato-Oncology, University of Milan, 20141 Milan, Italy

**Keywords:** locally advanced NSCLC, T4 tumor, superior vena cava involvement, prosthesis

## Abstract

**Simple Summary:**

The superior vena cava (SVC)’s involvement in non-small-cell lung cancer (NSCLC) has been considered a technical and oncological contraindication for surgery. In recent decades, different studies have demonstrated that surgery should not be contraindicated per se, but in highly selected cases and specialized centers, it could be curative with acceptable risks. Nevertheless, the tangential resection of the SVC or patch reconstruction have different surgical risks from prosthetic replacement. Moreover, the percentage of SVC involvement may influence the prognoses of these selected patients. Our intention was to investigate the relation between the rate of SVC involvement and surgical and oncological outcomes. The conclusions of our retrospective study may improve the management of patients with T4 NSCLC and SVC invasion.

**Abstract:**

Background: Surgery for non-small-cell lung cancers (NSCLCs) invading the superior vena cava (SVC) is rarely performed due to surgical complexities and reported poor prognoses. Different methods have been described to reconstruct the SVC, such as direct suture, patch use or prosthesis, according to its circumferential involvement. The aim of our study was to analyze the short- and long-term results of different types of SVC resection and reconstruction for T4 NSCLCs. Methods: Between January 2000 and December 2019, 80 patients received an anatomical lung resection with SVC surgery in this multicenter retrospective study. The partial resection and direct suture or patch reconstruction group included 64 patients, while the complete resection and prosthesis reconstruction group included 16 patients. The primary endpoints were as follows: long-term survival and disease-free survival. The secondary endpoints were as follows: perioperative complications and 30- and 90-day mortality. Unpaired t-tests or Mann–Whitney U tests for non-parametric variables were applied to discrete or continuous data, and the chi-square test was applied to dichotomous or categorical data. Survival rates were calculated using the Kaplan–Meier method and compared using the log-rank test. Results: No differences were found between the two groups in terms of general characteristics and surgical, oncological and survival outcomes. In particular, there were no differences in terms of early (50.0% vs. 68.8%, *p* = 0.178) and late complication frequency (12.5% vs. 12.5%, *p* = 1.000), 30- and 90-day mortality, R status, recurrence, overall survival (33.89 ± 40.35 vs. 35.70 ± 51.43 months, *p* = 0.432) and disease-free survival (27.56 ± 40.36 vs. 31.28 ± 53.08 months, *p* = 0.668). The multivariate analysis demonstrated that age was the only independent predictive factor for overall survival. Conclusions: According to our results, SVC resection has good oncological and survival outcomes, regardless of the proportion of circumferential involvement and the type of reconstruction.

## 1. Introduction

Non-small-cell lung cancer (NSCLC) continues to be the most common cancer worldwide and is frequently diagnosed at an advanced stage [1]. NSCLC that invades the mediastinum and the great vessels is considered a T4 disease [2]. Surgery is rarely performed when the superior vena cava (SVC) is involved due to surgical complexities and reported poor prognoses. Moreover, randomized studies are scarce and what we know of these patients is derived from retrospective single-center or multicenter studies [3,4,5,6,7,8]. According to the literature [8], surgery should not be contraindicated per se, but in highly selected cases and specialized centers, it could be curative with acceptable risks. Nevertheless, which patients may benefit from the surgical resection has not been well-established, and the reality may be different according to the percentage of circumferential involvement of the vessel. In fact, the tangential resection of the SVC or patch reconstruction have different risks from prosthetic replacement, which requires a demanding and fast procedure with different postoperative management.

We describe our experience with the surgical resection of NSCLC invading the SVC and compare the short- and long-term surgical outcomes of these highly selected cases with the intent to better establish which patients should be encouraged to have surgery.

## 2. Materials and Methods

Patients with SVC involvements for NSCLC at four international high-volume centers (>1000 major surgeries per year and large experience in locally advanced cases) between January 2000 and December 2019 were retrospectively analyzed. The paper was written according to the STROCSS (strengthening the reporting of cohort studies in surgery) criteria, and the checklist is provided as Appendix A [9]. The institutional review boards waived the need for ethical approval and consent for the retrospectively obtained and anonymized data.

Patients with T4 NSCLC and SVC involvement were included in the study. The surgical procedures have already been described in detail in the literature [10]. In the case of a tumor invasion of less than 25% of the SVC circumference, a tangential resection with a direct repair using a Satinsky side clamp was performed. Invasions between 25% and 50% were reconstructed with a patch of autologous pericardium. In the case of a tumor invasion of more than 50% of the SVC, a circumferential resection of the SVC and a total replacement with a ringed polytetrafluoroethylene (PTFE) graft or custom-made bovine pericardial tube was performed.

Patients were divided into two groups. Group A included patients who had a circumferential involvement of the SVC of less than 50%, and resection was followed by direct suture or patch reconstruction; meanwhile, Group B included patients whose SVC was replaced due to involvement >50%.

Postoperatively, no anticoagulation therapy was prescribed in patients who underwent a partial SVC resection, while in patients with a PTFE prosthesis, anticoagulation therapy using low-molecular-weight heparin was prescribed for 1 month and maintained afterward with oral anticoagulants. Patients with a bovine pericardium prosthesis received 150mg of ticlopidine daily after low-molecular-weight heparin.

The primary endpoints of this retrospective study were long-term survival (all-cause survival) and disease-free survival (DFS). The secondary endpoints were as follows: perioperative complications (rates and types) and 30- and 90-day mortality.

Patient demographics, tumor characteristics, surgical and oncological therapy, postoperative complications (atrial fibrillation, atelectasis, hemothorax, anemia, prolonged air leak (PAL), bronchopleural fistula (BPF), acute renal failure, stroke, SVC thrombosis, morbidity and mortality were analyzed. Overall survival (OS) was considered from surgery until the date of the last event, death for any cause or the last contact with the patient.

During the lengthy study period, neoadjuvant and adjuvant therapies evolved but consisted of platinum-based regimens and a median dose of 60 Gy for radiation therapy. Upfront surgery (surgery as first-line therapy) was considered and adjuvant chemotherapy, with or without radiotherapy, was performed according to the pathological lymph node status in clinical N0-1 disease. Induction chemotherapy followed by surgery was performed in cN2 disease with a radiological response, and adjuvant therapy was administered after a multidisciplinary postoperative evaluation. Adjuvant radiotherapy was administered in patients with positive margins (R1-2) or N2 bulky disease, usually in a daily dose of 1.8 Gy up to a total dose of 55–60 Gy.

Postoperative surveillance involved a physical examination, chest CT scan and upper abdominal ultrasound examination, or a total body CT scan, which were performed every 6 months for the first two years after surgery and at one-year intervals thereafter. If symptoms of disease recurrence occurred, additional examinations were performed regardless of the regular follow-up schedule.

TNM Staging was based on the 8th edition of the *AJCC Cancer Staging Manual* [2].

### Statistical Analysis

Statistical analyses were conducted using IBM Corp., released in 2017 (IBM SPSS Statistics for Windows, Version 25.0. Armonk, NY, USA: IBM Corp). Continuous variables were expressed as mean ± standard deviation (SD) or median and range. Categorical variables were expressed as numbers and percentages. All results were considered significant at *p* < 0.05. Outcomes were assessed using the Mann–Whitney U test for continuous measures and Pearson’s χ2 test for discrete variables. A Cox proportional hazards model was constructed based on hypothesized clinical relevance and univariate analysis results (*p* < 0.2). Survival rates were calculated using the Kaplan–Meier method and compared with the log-rank test.

## 3. Results

Eighty patients received an anatomical resection with an SVC resection and reconstruction. Group A included 64 patients, and 42 patients (65.6%) underwent a direct suture of the SVC while 22 (34.4%) underwent patch reconstruction. Group B included 16 patients.

### 3.1. Perioperative Results

The general and perioperative characteristics of the patients are shown in Table 1.

There were no differences between the two groups regarding age, gender, smoking habits, lung functions, Charlson comorbidity index and induction therapy (all patients had a stable disease according to RECIST Criteria 1.1 [11]). 

No differences were found in the surgical approach (thoracotomy was the preferred approach in both groups (90.6 vs. 87.5%)), intraoperative complications (no statistical differences, but the two complications that did happen occurred in Group A, *p* = 0.774), resection of other structures, intensive care unit (ICU) stay as frequency (81.3 vs. 75.0%, *p* = 0.576) or duration (*p* = 0.264), length of hospital stay (LOS) (13.83 ± 12.10 vs. 14.81 ± 7.39 days, *p* = 0.757), or 30- (7.8 vs. 6.3%, *p* = 0.832) and 90-day mortality (14.1 vs. 12.5%, *p* = 0.871). 

Early and long-term complications are shown in Table 2. No differences were found in the frequency of early complications (50.0% vs. 68.8%, *p* = 0.178), frequency of late complications (12.5% vs. 12.5%, *p* = 1.000), type of early and late complication (*p* = 0.864 and 0.879, respectively) or surgery for late complications (25.0% vs. 31.3%, *p* = 0.107). 

### 3.2. Oncological Outcomes and Long-Term Survival

The pathological and survival results are shown in Table 3.

No differences were found in the histology (in Group A, adenocarcinoma and squamous-cell carcinoma had the same frequency, occurring in 45.3% of patients; meanwhile, adenocarcinoma was the most common cancer in Group B, occurring in 62.5% of patients, *p* = 0.496), staging, R status (microscopic margins were positive in three patients of Group A and 1 patient of Group B, *p* = 0.798), adjuvant treatment, recurrence (54.7% vs. 43.8%, *p* = 0.433), and treatment of recurrence (*p* = 0.352) of the groups. 

Seventeen patients (26.6%) of Group A and two patients (12.5%) of Group B are currently alive (*p* = 0.237). There were 11 deaths (17.2%) for non-oncological reasons in Group A and 4 (25.0%) in Group B (*p* = 0.456).

The mean overall survival (OS) was 33.89 ± 40.35 months in Group A and 35.70 ± 51.43 months in Group B (*p* = 0.432, Figure 1).

The median disease-free survival (DFS) was 27.56 ± 40.36 months in Group A and 31.28 ± 53.08 in Group B (*p* = 0.668, Figure 2).

One-, three- and five-year OS rates were 65.6%, 28.1% and 18.8% in Group A, respectively, and 68.8%, 18.8% and 18.8% in Group B, respectively (*p* = 0.813, 0.446 and 1.000, respectively). One-, three- and five-year DFS rates were 46.9%, 23.4% and 14.1% in Group A, respectively, and 56.3%, 18.8% and 18.8% in Group B, respectively (*p* = 0.502, 0.688 and 0.639, respectively).

During multivariate analyses, age was found to be the only independent predictive factor for OS (Table 4).

## 4. Discussion

T4 lung cancers are a heterogeneous group with various types of infiltration, including SVC involvement [12]. Radical en bloc resection remains the gold standard in improving survival in patients with NSCLC. In the past, SVC involvement has been considered a technical and oncological contraindication for surgery [13]. However, with advancements in thoracic and cardiovascular surgery, the limits of surgical resection in locally advanced NSCLCs are beginning to shift. Less than 250 cases of SVC resection in lung cancer have been reported in the recent literature, with the largest to date involving 52 patients [14]. Most of the studies analyze patients treated in the 1980s, without thorough mediastinal staging, who received now out-of-date oncological treatments for mediastinal and lung cancers that require different treatment strategies and prognoses. Moreover, some of them also included SVC resection for N2-bulky disease, consequently resulting in discouraging long-term survival rates. In these studies, the reported 5 yr OS was 15–29% [10,15].

The evolution of the knowledge of tumor biology and adjuvant therapies has raised reported long-term survival from 15% to 40% [8,10].

However, despite improvements in surgical and anesthesiologic management, the morbidity and mortality of these extended surgical resections remain significantly higher than standard lung resections, ranging from 10–14% in recent studies [3,8].

Therefore, the careful selection of the patients in a multidisciplinary team discussion, an experienced anesthetist and a surgical team are mandatory to achieve improved prognoses in patients [16,17].

Our study reflects the results of SVC resections in four international centers in the last 20 years. Patients with N2-disease underwent induction therapy after a multidisciplinary discussion. We separately analyzed partial or complete resection of the SVC to find out any differences in morbidity, mortality and prognosis.

One of the primary endpoints of our study was to evaluate differences in OS between our groups. The survival rates of our two groups were not statistically different in terms of OS, one-, three- and five-year survival rates being 33.89 ± 40.35 months, 65.6%, 28.1% and 18.8%, respectively in Group A and 35.70 ± 51.43 months, 68.8%, 18.8% and 18.8%, respectively in Group B. Comparing our results with previous studies, our mid- and long-term OS rates were lower than the literature. In fact, Shargall et al. [3] reported a 3- year OS rate of 57% and Chenesseau et al. [8] reported a 3- and 5-year OS rate of 45 and 50%, respectively. In his first retrospective during the 1980s, Spaggiari et al. reported a 5-year survival rate of 29% [18]. We believe these poor long-term results could firstly be explained by the fact that most of our patients did not undergo an adjuvant treatment. In fact, only 55% of partial resection and 38% of graft replacement patients had postoperative chemotherapy and/or radiotherapy. This was due to preoperative comorbidities and low performance status after the intervention. Moreover, 17% of Group A and 25% of Group B died for non-oncological reasons. This great proportion of deaths undoubtedly had an impact on long-term survival rate.

The other primary endpoint of the study was DFS, which was not statistically different between the two groups (27.56 ± 40.36 vs. 31.28 ± 53.08, *p* = 0.668). Furthermore, median DFS was long: 28 months in Group A and 31 months in Group B. In fact, 1 and 3-year DFS rates were 47, 23% and 56.19% for partial and total resections, respectively. These results are perfectly in line with the same studies cited above which report a better OS, even if very few of them analyzed DFS rate [3,8]. In fact, Shargall et al. [3] reported a DFS of 53 and 27% at 1 and 3 years, respectively.

These considerations demonstrate that our poor long-term outcomes are due to the lack of postoperative treatment and underscore the importance of this oncological step. Unfortunately, precise data on postoperative treatment are lacking in almost all studies, so a precise comparison cannot be conducted.

The secondary endpoints of our study were perioperative complications and 30- and 90-day mortality. No differences were found in intraoperative complications, ICU stay, LOS, and 30- and 90-day mortality. Most of the patients experiencing complications had a low Clavien–Dindo score. Most of the complications were managed with pharmacological treatments or endoscopic interventions. In fact, the most frequent early complications were atrial fibrillation (14.1% in Group A and 31.3% in Group B) and atelectasis (15.7% in Group A and 18.8% in Group B). Surprisingly, Group B patients did not experience more major complications than Group A. In fact, even if SVC partial resection is thought to be less invasive, OR readmission for early complications were experienced almost only by patients of Group A (7 vs 1 patients, *p* = 0.576). Similarly, SVC thrombosis happened solely in two patients who had a partial resection. Moreover, no differences were found in 30- (7.8 vs. 6.3%, *p* = 0.832) and 90-day mortality (14.1 vs. 12.5%, *p* = 0.871).

Shargall et al. [3] reported a morbidity rate of 26% and a 30-day mortality of 14%. In their 30 years of institutional analysis of SVC replacements, Chenesseau et al. [8] reported no intraoperative deaths and a morbidity and mortality rate of 27% and 10%, respectively. Surgery for early complications was performed only for two patients with cardiac herniation. 

One of the most feared complications in SVC replacement is graft thrombosis, which is a very rare complication as reported in the literature, and almost all cases are treated conservatively with anticoagulation [3,6,8]. Graft patency does not seem to be an immediate issue but a matter of long-term concern. In fact, as analyzed by Oizumi et al. [6] in a series of patients without postoperative anticoagulation, SVC thrombosis is well-managed medically or with a Fogarty balloon. The high graft thrombosis rate they reported (13.6%) shows the importance of postoperative routine anticoagulation.

We think the only technical issue of the patch repair or graft substitution of SVC clamping is intraoperative management and time. What is important is the reconstruction result: to maintain an adequate lumen that is as large as possible to ensure a higher flow rate with lesser resistance. The theory that a prosthetic replacement yields a larger thrombogenic surface does not seem to be demonstrated in our study. 

Another important issue is the resection status. The rate of positive margins was very low compared to the literature [8,10] and did not statistically differ between the two groups (4.7% vs. 6.3%). In fact, the reported R1 rate is higher and ranges from 12 to 18% [8,10]. We think it is an entirely acceptable rate considering the surgical complexities and the challenging situation surgeons sometimes have to face in locally advanced tumors, especially in proximal (near cervical) or distal (atrial) extensions of cancer. An accurate preoperative selection of patients and a thorough intraoperative decision of the reconstruction technique may restrict the R+ rate in those limits.

The study limitations are as follows. This is a retrospective study, involving multiple centers which may lead to selection bias. Results may be influenced by tumor characteristics, surgical technique and individual surgeon experience and preferences. The sample size is small due to the limited number of patients that present with resectable SVC involvement. Moreover, we do not have data of patients with SVC involvement treated differently from surgery. The staging system during the period taken into account has changed, thus stage IIIB patients (clinical or pathological N2) were included in the study. Finally, disease recurrence may be considered a surrogate endpoint due to the small sample size; thus, larger groups of randomized trials are required, although the number of operable NSCLCs with SVC invasion is very limited, making this research unlikely [19,20].

## 5. Conclusions

According to our results, SVC resection has good oncological and survival outcomes, regardless of the proportion of circumferential involvement and the type of reconstruction.

## Figures and Tables

**Figure 1 cancers-14-06138-f001:**
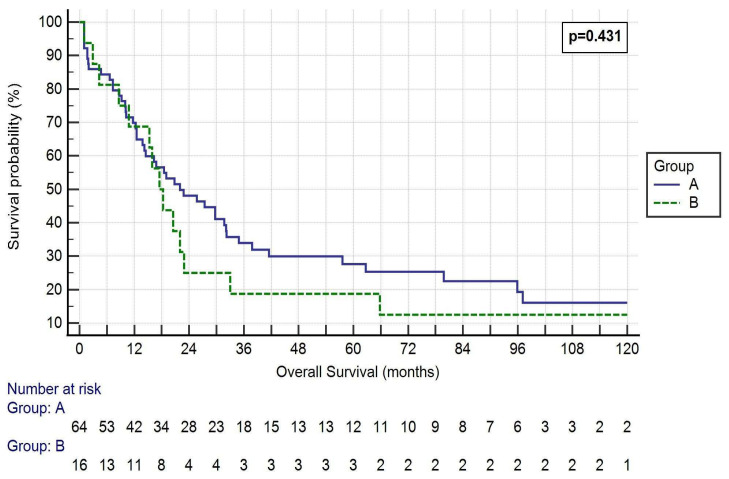
Kaplan–Meier curve of the OS of the two Groups. Log-rank *p* = 0.431. Group A: <50% SVC involvement and resection followed by direct suture or patch reconstruction; Group B: >50% SVC involvement and prosthesis reconstruction.

**Figure 2 cancers-14-06138-f002:**
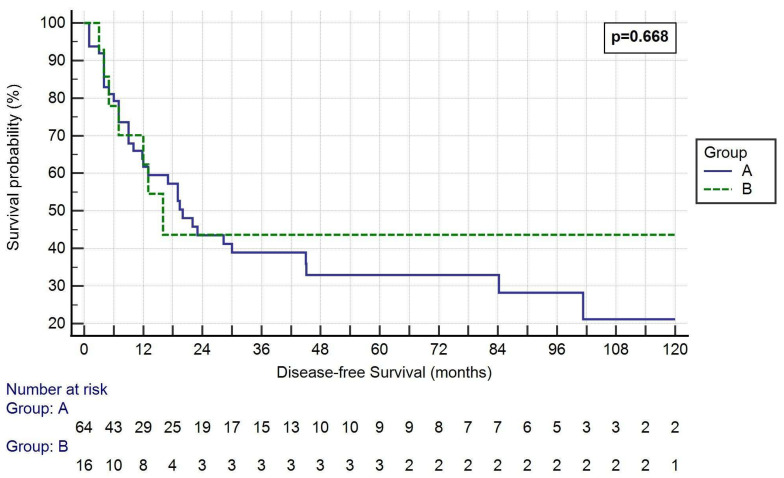
Kaplan–Meier curve of the DFS of the two groups. Log-rank *p* = 0.668. Group A: <50% SVC involvement and resection followed by direct suture or patch reconstruction; Group B: >50% SVC involvement and prosthesis reconstruction.

**Table 1 cancers-14-06138-t001:** General and perioperative characteristics of patients.

	Group A (n = 64)	Group B (n = 16)	*p*-Value
Age	62.5(54.5–69.75)	64.0(54.25–70.25)	0.574
Sex Male Female	53 (82.8%) 11 (17.2%)	11 (68.8%) 5 (31.3%)	0.208
Smoking	48 (75.0%	13 (81.3%)	0.599
FEV1 (%)	85.0 (76.25–94.0)	73.0 (57.75–93.5)	0.105
DLCO (%)	78.0 (63.5–92.5)	71.5 (58.0–93.5)	0.476
Charlson comorbidity index	5.0 (4.0–6.0)	5.0 (4.25–6.0)	0.827
cTNM staging (8th edition) IIIA IIIB	43 (67.2%) 21 (32.8%)	12 (75.0%) 4 (25.0%)	0.546
Induction therapy CT CT + RT	27 (42.2%) 6 (9.4%)	4 (25.0%) 2 (12.5%)	0.451
Surgical access Thoracotomy Sternotomy Hemiclamshell Clamshell	58 (90.6%) 0 (0%) 3 (4.7%) 3 (4.7%)	14 (87.5%) 1 (6.3%) 0 (0%) 1 (6.3%)	0.185
Lung resection Lobectomy Bilobectomy Pneumonectomy Sleeve lobectomy Sleeve pneumonectomy	15 (23.4%) 2 (3.1%) 23 (35.9%) 24 (37.5%) 0 (0%)	2 (12.5%) 2 (12.5%) 7 (43.8%) 4 (25.0%) 1 (6.3%)	0.101
Intraoperative complications Bleeding Ventricular fibrillation	1 (1.6%) 1 (1.6%)	0 (0%) 0 (0%)	0.774
Main pulmonary artery infiltration	14 (21.9%)	6 (37.5%)	0.197
Right atrium infiltration	1 (1.6%)	1 (6.3%)	0.283
CPB use (pts)	1 (1.6%)	0 (0%)	0.615
ICU stay (pts)	52 (81.3%)	12 (75.0%)	0.576
ICU stay (days)	1.0(1.0–3.0)	1.0(0.0–1.75)	0.264
LOH stay (days)	13.83(12.10)	14.81(7.39)	0.757
30-day mortality	5 (7.8%)	1 (6.3%)	0.832
90-day mortality	9 (14.1%)	2 (12.5%)	0.871
Cause of 90-day mortality Pulmonary embolism Myocardial Infarction ARDS PEA Unknown cause (patient found dead outside of the hospital)	2 (22.2%) 2 (22.2%) 2 (22.2%) 1 (11.1%) 2 (22.2%)	1 (50.0%) 1 (50.0%) 0 (0%) 0 (0%) 0 (0%)	0.729

Notes: Data are presented as mean (SD), median (P25–P75) or n (%). Abbreviations: Group A: <50% SVC involvement and resection followed by direct suture or patch reconstruction; Group B: >50% SVC involvement and prosthesis reconstruction; pts: patients; FEV1, forced expiratory volume in 1 s; DLCO, diffusing capacity for carbon monoxide; CT: chemotherapy; RT: radiotherapy; CPB: cardiopulmonary bypass; ICU: intensive care unit; LOH: length of hospital; ARDS: acute respiratory distress syndrome; PEA: pulseless electrical activity.

**Table 2 cancers-14-06138-t002:** Early and long-term complications.

	Group A (*n* = 64)	Group B (*n* = 16)	*p*-Value
Early complications (pts)	32 (50.0%)	11 (68.8%)	0.178
More than one complication	7 (10.9%)	0 (0%)	0.166
Early complications (type) Atrial fibrillation Atelectasis Haemothorax Anaemia PAL BPF Acute renal failure Stroke SVC thrombosis	9 (14.1%) 10 (15.7%) 3 (4.7%) 2 (3.1%) 1 (1.6%) 2 (3.1%) 2 (3.1%) 1 (1.6%) 2 (3.1%)	5 (31.3%) 3 (18.8%) 2 (12.5%) 1 (6.3%) 0 (0%) 0 (0%) 0 (0%) 0 (0%) 0 (0%)	0.864
Surgery for early complications	7 (10.9%)	1 (6.3%)	0.576
Clavien–Dindo Classification Grade 1 Grade 2 Grade 3A Grade 3B Grade IVA	1 (1.6%) 11 (17.2%) 11 (17.2%) 7 (10.9%) 2 (3.1%)	0 (0%) 7 (43.8%) 3 (18.8%) 1 (6.3%) 0 (0%)	0.303
Polytrasfusion	4 (6.3%)	2 (12.5%)	0.396
Late complications (pts)	8 (12.5%)	2 (12.5%)	1.000
Late complications (type) cardiac complications BPF Respiratory complications SVC thrombosis Bronchial stenosis PE Pleural effusion	1 (1.6%) 2 (3.1%) 1 (1.6%) 1 (1.6%) 1 (1.6%) 1 (1.6%) 1 (1.6%)	1 (6.3%) 0 (0%) 1 (6.3%) 0 (0%) 0 (0%) 0 (0%) 0 (0%)	0.879
Surgery for late complications Fenestration	2 (3.1%)	0 (0%)	0.474

Notes: Data are presented as mean (SD), median (P25–P75) or n (%). Abbreviations: Group A: <50% SVC involvement and resection followed by direct suture or patch reconstruction; Group B: >50% SVC involvement and prosthesis reconstruction; pts: patients; PAL: prolonged air leak; BPF: bronchopleural fistula; SVC: superior vena cava; PE: pulmonary embolism.

**Table 3 cancers-14-06138-t003:** Histological and oncological results.

	Group A (*n* = 64)	Group B (*n* = 16)	*p*-Value
Histology Adenocarcinoma SSC Large-cell carcinoma Adenosquamous	29 (45.3%) 29 (45.3%) 2 (3.1%) 4 (6.3%)	10 (62.5%) 4 (25.0%) 1 (6.3%) 1 (6.3%)	0.496
pTNM staging (8th edition) IIIA IIIB	37 (57.8%) 27 (42.2%)	11 (68.8%) 5 (31.3%)	0.424
pN staging N0 N1 N2	12 (18.8%) 24 (37.5%) 28 (43.8%)	2 (25.0%) 7 (43.9%) 5 (31.3%)	0.649
R status (R1)	3 (4.7%)	1 (6.3%)	0.798
Adjuvant therapy CT RT CT + RT	12 (18.8%) 13 (20.3%) 4 (6.3%)	2 (12.5%) 1 (6.3%) 3 (18.8%)	0.245
Recurrence (pts)	35 (54.7%)	7 (43.8%)	0.433
Pattern of recurrence Local Distant Local + distant	15 (23.4%) 14 (21.9%) 6 (9.4%)	5 (31.3%) 2 (12.5%) 0 (0%)	0.441
Number of recurrences One Two Three	25 (39.1%) 8 (12.5%) 2 (3.1%)	7 (43.8%) 0 (0%) 0 (0%)	0.408
Treatment of recurrence RT CT Surgery Surgery + RT CT + RT	9 (14.1%) 14 (21.9%) 1 (1.6%) 1 (1.6%) 7 (10.9%)	3 (18.8%) 1 (6.3%) 0 (0%) 0 (0%) 0 (0%)	0.352
Deaths	47 (73.4%)	14 (87.5%)	0.237
Deaths for non-oncological reasons	11 (17.2%)	4 (25.0%)	0.456
1-year survival	42 (65.6%)	11 (68.8%)	0.813
3-year survival	18 (28.1%)	3 (18.8%)	0.446
Survival > 5 years	12 (18.8%)	3 (18.8%)	1.000
DFS at 1 year	30 (46.9%)	9 (56.3%)	0.502
DFS at 3 years	15 (23.4%)	3 (18.8%)	0.688
DFS at 5 years	9 (14.1%)	3 (18.8%)	0.639

Notes: Data are presented as mean (SD), median (P25–P75) or n (%). Abbreviations: Group A: <50% SVC involvement and resection followed by direct suture or patch reconstruction; Group B: >50% SVC involvement and prosthesis reconstruction; SSC: squamous-cell carcinoma; R: residual tumor; CT: chemotherapy; RT: radiotherapy; pts: patients; DFS: disease-free survival.

**Table 4 cancers-14-06138-t004:** Univariate and multivariate analyses with overall survival in months as the dependent variable.

	Univariate Analysis	Multivariate Analysis		
Variable	*p*-Value	HR	95% CI	*p*-Value
Age	0.273	5.570	1.585–19.577	0.007 *
Gender	0.176	-	-	-
Pneumonectomy	0.302	-	-	-
Induction therapy	0.742	-	-	-
Lymph node status	0.361	-	-	-
Resection status	0.514	-	-	-
Adjuvant therapy	0.297	-	-	-

Notes: * *p* < 0.05.

## Data Availability

The data presented in this study are available on request from the corresponding author.

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
