# Peer review of "A Multicenter Retrospective Cohort Study on Superior Vena Cava Resection in Non-Small-Cell Lung Cancer Surgery"

_cancers, 2022, doi:10.3390/cancers14246138_

Round 1

Reviewer 1 Report

Review or multicenter retrospective cohort study on superior vena cava resection in NSCLC surgery

The purpose of this study is a retrospective review of lungs with superior vena cava surgical resection of long- and short-term outcomes in patients with NSCLC.

Abstract:

Between 2000-2019 80 patients received an anatomical lung resection with SVC surgery in this multicenter retrospective study. Partial resection and direct suture or patch reconstruction group (N=64) and complete resection and prosthesis reconstruction (N=16). A sentence stating the methods of analysis for comparison of short and long term survival should be stated in the abstract.

Introduction:

The introduction should be briefly extended with 1-2 more paragraphs delineating the background for the study. Why should this study be performed? I know the paper states there is not much previous literature but at least 2-3 recent relevant studies should be listed. What did these previous studies find and why was this retrospective study done? What is the approximate incidence of NSCLC with SVC involvement?

Materials and Methods: This section needs a lot of restructuring, specifics, and rewriting.

1.       Please give additional detail on how the 4 international high-volume centers were selected for study.

2.       Were patients placed in 2 groups because of the low number of cases? This should be stated in the methods. ‘Group A’ and ‘Group B’ should be converted to a more clinically relevant nomenclature unless you put how these groups were separated at the bottom of all tables and figures. Figures and tables should always stand-alone and the reader should not need to refer to the text for interpretation.

3.       Please convert all bulleted text to sentences. Is long term survival all-cause survival? Please state

4.       Please list all perioperative complications and separate ‘Outcomes’ into their own subsection for ease of reading.

5.       5. Line 88 page 2-how and why were patient demographics, tumor characteristics, etc analyzed and for what purposes? Some of these are outcomes and some of these should be covariates for the multivariable models.

6.       On page 2 can you please explain what is meant by “upfront surgery” (line 92).

7.       Please state the covariates assessed between the two groups that appear in tables 1-2 in this section and clarify the purpose of the paper.

Statistical analysis

1.       For the Cox proportional hazard analysis was the proportional hazards assumptions tested and met?

2.       Based on the objective of the study as stated there should be an assessment for confounding variables between Group A and B not a predictive model for survival differences. If the objective of the study is to distinguish covariates predictive for survival between the two groups then that needs to be updated in the objective statement and the methods of the paper. If univariate variable selection was to assess differences between the two groups and outcomes (as a confounder) please state that.

3.       Was there a minimum number of outcomes for assessment?

Results

1.       This section should be divided into paragraphs and extended some to review the differences between the 2 groups A vs. B with respect to all the output in the tables. Otherwise, the results as outlined in the table are well structured.

2.       The KM plot with life table is clear and easy to understand.

Discussion

1.       The discussion needs to be restructured to support the objectives of the paper and the results. This section as written is incomplete and difficult to read and doesn’t do enough with the results of the study.

Author Response

Review or multicenter retrospective cohort study on superior vena cava resection in NSCLC surgery

The purpose of this study is a retrospective review of lungs with superior vena cava surgical resection of long- and short-term outcomes in patients with NSCLC.

Abstract:

Between 2000-2019 80 patients received an anatomical lung resection with SVC surgery in this multicenter retrospective study. Partial resection and direct suture or patch reconstruction group (N=64) and complete resection and prosthesis reconstruction (N=16). A sentence stating the methods of analysis for comparison of short and long term survival should be stated in the abstract.

Reply: thank you for your comment. We added these sentences in the methods section of the abstract :”Primary endpoints were: long-term survival and disease recurrence; secondary endpoints: perioperative complications and 30-and 90-day mortality. Unpaired t test or Mann-Whitney U test for non parametric variables were applied to discrete or continuous data, and the chi-square test was applied to dichotomous or categorical data. Survival rates were calculated by using the Kaplan-Meier method and compared with the Log-rank test.”

Introduction:

The introduction should be briefly extended with 1-2 more paragraphs delineating the background for the study. Why should this study be performed? I know the paper states there is not much previous literature but at least 2-3 recent relevant studies should be listed. What did these previous studies find and why was this retrospective study done? What is the approximate incidence of NSCLC with SVC involvement?

Reply: thank you. We added two paragraphs, one discussing surgical resection (“... it is not well established which patients may benefit from the surgical resection, that may be different according to the percentage of the circumferential involvement of the vessel. In fact,...” and the other about the reason for this article (“...with the intent to better establish which patients should be encouraged to have surgery….”). 

Materials and Methods: This section needs a lot of restructuring, specifics, and rewriting.

  1.       Please give additional detail on how the 4 international high-volume centers were selected for study.

Reply: thank you. One of the authors worked in three of the four centers and with all of the authors of the article. The authors have been collaborating in T4 NSCLC research for more than 3 years. There are no other reasons for this selection other than scientific connections and volume and experience in locally advanced NSCLC surgery. 

  1.       Were patients placed in 2 groups because of the low number of cases? This should be stated in the methods. ‘Group A’ and ‘Group B’ should be converted to a more clinically relevant nomenclature unless you put how these groups were separated at the bottom of all tables and figures. Figures and tables should always stand-alone and the reader should not need to refer to the text for interpretation.

Reply: thank you for your valuable comment. We added this sentence at the bottom of the tables and figures: “Group A: <50% SVC involvement and resection followed by direct suture or patch reconstruction; Group B: >50% SVC involvement and prosthesis reconstruction.”

  1.       Please convert all bulleted text to sentences. Is long term survival all-cause survival? Please state

Reply: thank you. We did it.

  1.       Please list all perioperative complications and separate ‘Outcomes’ into their own subsection for ease of reading.

Reply: thank you for your comment. We listed the perioperative complications in the materials and methods section, added the significance of Overall survival and created a single paragraph with endpoints. 

  1.       5. Line 88 page 2-how and why were patient demographics, tumor characteristics, etc analyzed and for what purposes? Some of these are outcomes and some of these should be covariates for the multivariable models.

Reply: we do not understand your comment. Some of those variables are used for multivariate analysis and that is just the list of the variables considered in the study. We may extend the description. 

  1.       On page 2 can you please explain what is meant by “upfront surgery” (line 92).

Reply: thank you. Upfront surgery is an already established term for surgery with no induction therapy for locally advanced cancer. We added this sentence: “surgery as first-line therapy”

  1.       Please state the covariates assessed between the two groups that appear in tables 1-2 in this section and clarify the purpose of the paper.

Reply: thank you for your comment. We believe that it would be redundant to list all of the covariates in the materials and methods. The purpose of the article was added in the introduction section.

Statistical analysis

  1.       For the Cox proportional hazard analysis was the proportional hazards assumptions tested and met?

Reply: thank you for your comment. It was tested and met.

  1.       Based on the objective of the study as stated there should be an assessment for confounding variables between Group A and B not a predictive model for survival differences. If the objective of the study is to distinguish covariates predictive for survival between the two groups then that needs to be updated in the objective statement and the methods of the paper. If univariate variable selection was to assess differences between the two groups and outcomes (as a confounder) please state that.

Reply: thank you. The idea of the study is to compare the two groups for all the aspects, not just to distinguish covariates predictive for survival. The univariate analysis was done just to select covariates for multivariate analysis.

  1.       Was there a minimum number of outcomes for assessment?

Reply: thank you for your comment. No, there was not.

Results

  1.       This section should be divided into paragraphs and extended some to review the differences between the 2 groups A vs. B with respect to all the output in the tables. Otherwise, the results as outlined in the table are well structured.

Reply: thank you for your comment. We extended the results section and divided into paragraphs.

  1.       The KM plot with life table is clear and easy to understand.

Reply: thank you.

Discussion

  1.       The discussion needs to be restructured to support the objectives of the paper and the results. This section as written is incomplete and difficult to read and doesn’t do enough with the results of the study.

Reply: Thank you for your valuable comment. We reconstructed the discussion, focusing firstly on primary endpoints and trying to compare our results with literature. 

Reviewer 2 Report

The paper analyzes a particular situation regarding the operability of advanced NSCLC, specifically the infiltration of the vena cava.

This work certainly represents a useful addition to the literature since I don't know of any multicentric studies on this topic.

The introduction is brief but contains the essential background information on the work.

Materials and methods are clearly exposed and the results presented adequately, thanks also to the numerous tables.

In the discussion, the works in the literature were correctly cited, correlating them with the results obtained.

I have just one question for the authors: in the manuscript you mention neoadjuvant chemotherapy only. In the experience of one of the centers involved in the study, have there been any cases of advanced NSCLC with invasion of the cava treated with neoadjuvant immunotherapy? If yes, in your opinion can this be a technical obstacle for an easy resection of the vena cava?

Author Response

 We thank you for your comments. Unfortunately, we did not had a patient treated with neoadjuvant immunotherapy. We are sure there will be considering the amazing results with immunotherapy. Nevertheless, the desmoplastic reaction after immuno may increase difficulty of the procedure. In a previous article published by surgeons from Shanghai Chest Hospital, they mention this aspect after immuno, which may increase the operative time and overall blood loss. Although, our centers are high volume centers, therefore our surgeons have a lot of experience operating on patients with fibrosis and desmoplastic changes of the tissue.

Reviewer 3 Report

Reviewer

Initial comments

This issue is of great importance, as it deals with the therapeutic results of an aggressive disease and that, in fact, there is a need for differentiated treatment.

 TITLE:

A multicenter retrospective cohort study on superior vena cava resection in NSCLC surgery.

Comment:

Please, spell out NSCLC......

NSCLC...... non-small cell lung cancer

Simple Summary:

Comment:

It is suitable

Abstract:

Comment:

Lines 27-28….

Please, remove the space between these lines.

1. Introduction  

 Comment:

It is suitable

2. Materials and methods  

Comment:

It is suitable

2.1. Statistical Analysis

Comment:

It is suitable

3. Results

Comment:

It is suitable

3.1. Perioperative results

Comment:

It is suitable

3.2. Oncological outcomes and long-term survival

Comment:

It is suitable

4. Discussion

Comment:

It is suitable

5. Conclusions

Comment:

It is suitable

Supplementary Materials:

Comment:

It is suitable

References

Comment:

It is suitable

Thank you

Author Response

Initial comments

This issue is of great importance, as it deals with the therapeutic results of an aggressive disease and that, in fact, there is a need for differentiated treatment.

 TITLE:

 A multicenter retrospective cohort study on superior vena cava resection in NSCLC surgery.

Comment:

Please, spell out NSCLC......

NSCLC...... non-small cell lung cancer

Reply: thank you. We did it.

Simple Summary:

 Comment:

 It is suitable

Abstract:

Comment:

Lines 27-28….

Please, remove the space between these lines.

Reply: thank you. In the word manuscript there is no space. We tried to removed it.

  1. Introduction  

 Comment:

 It is suitable

  1. Materials and methods  

 Comment: 

It is suitable

2.1. Statistical Analysis

Comment:

It is suitable

  1. Results

Comment:

It is suitable

3.1. Perioperative results

 Comment:

 It is suitable

3.2. Oncological outcomes and long-term survival

 Comment:

It is suitable

  1. Discussion

 Comment:

 It is suitable

  1. Conclusions

 Comment:

 It is suitable

Supplementary Materials:

Comment:

It is suitable

References

 Comment: 

It is suitable

Thank you

Reply: thank you.

Round 2

Reviewer 1 Report

Thank you for responding to my comments questions. I appreciate the updates you have made and I think this paper is now fine for publication.